# Lutetium-177-Prostate-Specific Membrane Antigen Radioligand Therapy: What Is the Value of Post-Therapeutic Imaging?

**DOI:** 10.3390/biomedicines12071512

**Published:** 2024-07-08

**Authors:** Jules Zhang-Yin

**Affiliations:** Department of Nuclear Medicine, Clinique Sud Luxembourg, Vivalia, B-6700 Arlon, Belgium; juleszhangyin@gmail.com

**Keywords:** Lutetium-177, radioligand therapies, PSMA, personalized dosimetry, post-therapeutic imaging

## Abstract

Lutetium-177 (Lu-177)-labelled radioligand therapies (RLT) targeting prostate-specific membrane antigen (PSMA) present a promising treatment for patients with progressive metastasized castration-resistant prostate cancer (mCRPC). Personalized dosimetry, facilitated by post-therapeutic imaging, offers the potential to enhance treatment efficacy by customizing radiation doses to individual patient needs, thereby maximizing therapeutic benefits while minimizing toxicity to healthy tissues. However, implementing personalized dosimetry is resource-intensive, requiring multiple single-photon emission-computed tomography (SPECT)/CT scans and posing significant logistical challenges for both healthcare facilities and patients. Despite these challenges, personalized dosimetry can lead to optimized radiation delivery, improved safety, and better management of complex cases. Nevertheless, the financial and resource burdens complicate its adoption in routine clinical practice. While the European Association of Nuclear Medicine (EANM) supports personalized dosimetry, standardization is lacking due to these practical constraints. Further research and streamlined methodologies are essential to balance the benefits and feasibility of personalized dosimetry, potentially improving treatment outcomes for mCRPC patients.

## 1. Introduction

The success of the VISION study marks a significant milestone in the field of radionuclide ligand therapy (RLT), paving the way for a potential exponential increase in treatment options. The positive outcomes of this study strongly support the adoption of ^177^Lu-prostate-specific membrane antigen (PSMA)-617 as the standard treatment for advanced PSMA-positive mCRPC.

The VISION trial study was a pivotal phase III clinical study that assessed the efficacy and safety of PSMA-617 (Pluvicto^®^) in patients with mCRPC. It demonstrated significant improvements in both overall survival (OS) and radiographic progression-free survival (rPFS) for patients treated with PSMA-617 compared to those receiving standard care. Patients in the PSMA-617 group showed a marked increase in OS, and the treatment also prolonged rPFS, indicating a delay in disease progression. Additionally, higher prostate-specific antigen (PSA) response rates were observed in the PSMA-617 group, reflecting the therapy’s effectiveness. The treatment was generally well-tolerated, with manageable side effects such as fatigue, bone marrow toxicity, and mild renal impairment. These findings highlighted the substantial survival benefit and quality of life improvements for heavily pre-treated mCRPC patients. Overall, the VISION trial established PSMA-617 as an effective treatment option, confirming its clinical viability and leading to its approval for mCRPC treatment [1]. This trial was instrumental in leading to the Food and Drug Administration (FDA) and European Medicines Agency (EMA) approvals of PSMA-617 in 2022.

However, it is important to note that a non-negligible rate of grade III haematotoxicity has been observed. Additionally, there is considerable variability in absorbed doses among individual patients across different organs [1]. Therefore, conducting patient-specific dosimetry is crucial, a position also endorsed by the European Association of Nuclear Medicine (EANM) dosimetry committee [2]. The primary objective is to maximize the radiation dose to the tumor while minimizing the absorbed doses in normal tissues and organs. Generally, nuclear medicine therapies require adherence to dosimetry guidelines in accordance with international regulations (e.g., 2013/59/Euratom) [3], recommendations from the International Commission on Radiological Protection (ICRP 140) [4], and national legislations [2].

In routine clinical practice, obtaining patient-specific dosimetry often necessitates multiple iterative imaging procedures, which may not always be feasible. Conducting numerous single-photon emission-computed tomography/computed tomography (SPECT/CT) scans, in particular, presents logistical and resource challenges for both patients and healthcare facilities, potentially limiting patients’ access to this innovative treatment.

This review aims to explore the role of post-therapeutic imaging in patients undergoing ^177^Lu-PSMA-RLT, with a particular focus on personalized dosimetry. After presenting the current clinical practice ^177^Lu-PSMA-RLT, an examination of the state-of-the-art for personalized dosimetry for ^177^Lu-PSMA-RLT will be conducted, drawing comparisons with different methods. Finally, the advantages and disadvantages for post-therapeutic imaging will be discussed, considering the specific challenges and some proposals to overcome them.

## 2. The Present State-of-the-Art in Personalized Dosimetry for ^177^Lu-PSMA-RLT

### 2.1. General Procedures of the ^177^Lu-PSMA-RLT

#### 2.1.1. Indications

The primary PSMA ligands used in therapy are PSMA-1 (PSMA-I&T or ^177^Lu-PNT2002) and PSMA-617 (Pluvicto^®^). PSMA-617 received FDA approval in March 2022 and EMA approval in December 2022 for mCRPC patients progressing after treatments like enzalutamide (Xtandi^®^) or abiraterone acetate (Zytiga^®^) and taxane chemotherapy. This approval is based on the VISION phase III trial showing a 4-month survival benefit. The phase IIb TheraP trial suggests using ^177^Lu-PSMA-617 for patients treated with docetaxel but eligible for cabazitaxel. The 2023 EANM/SNMMI guidelines allow ^177^Lu-PSMA for mCRPC patients contraindicated for taxane treatments [5,6].

#### 2.1.2. Contraindications

Contraindications include poor overall health, acute infections, comorbidities affecting hydration, urinary issues, liver, and hematopoietic functions. The World Health Organization (WHO) grade ≥ 3 or life expectancy under 3 months disqualifies patients due to low quality of life benefits. Extensive support needs, psychiatric issues hindering isolation adherence, and specific radioprotection concerns (e.g., acute urine retention) also contraindicate use. Biological contraindications include Glomerular filtration rate (GFR) < 30 mL/min, high serum creatinine, significant hepatic cytolysis, leukopenia, neutropenia, and thrombocytopenia. Brain metastases, epiduritis, spinal cord compression, or pachymeningitis require a multidisciplinary approach. Those initially refused treatments may be reconsidered if conditions improve.

#### 2.1.3. Procedure

Treatment involves biological assessments, intravenous hydration before and after administration, and monitoring post-treatment.

Particularly, it is important to note that PSMA-targeted therapy is effective only if tumors express sufficient PSMA receptors. Evaluation of PSMA expression should be done using molecular imaging (preferred: ^68^Ga/^18^F-PSMA PET; alternatively: ^99^mTc-PSMA scintigraphy) alongside conventional imaging to identify PSMA-negative metastases. Simultaneous ^18^FDG-PET may not be mandatory for all but is helpful for evaluating uncertain lesions or suspected PSMA negativity, especially liver lesions. It is proposed to exclude patients with lesions ≥1 cm showing tumor uptake <0.5-fold of parotid uptake (or <1.0-fold liver uptake for ^68^Ga-PSMA-11) from ^177^Lu-PSMA-RLT (Figure 1).

The typical dose is 7.4 GBq per cycle, up to six cycles, with variations based on patient response. Post-treatment, whole-body scintigraphy, and/or SPECT-CT scans ensure proper administration. Patients receive detailed reports and hydration advice.

#### 2.1.4. Tolerance

^177^Lu-PSMA is well-tolerated; severe adverse effects are rare. Common side effects include fatigue, bone marrow toxicity, chronic xerostomia, renal insufficiency, nausea, vomiting, and headaches. Flare-up pain, particularly in bone metastases, may occur. Preventive treatments include ondansetron for vomiting and dexamethasone for other symptoms. High tumor volume may necessitate allopurinol. Blood counts are monitored bi-weekly, with possible interventions for hematopoietic involvement. Liver and kidney function tests are also performed. Treatment pauses or dose reductions may occur based on adverse effects or tolerance. Effective contraception is recommended for three months post-treatment due to potential fertility risks. Regular consultations with a nuclear medicine physician assess treatment efficacy and tolerance, with primary care physicians informed of progress to enhance management.

### 2.2. The State-of-the-Art of Dosimetry in Clinical Practice

#### 2.2.1. Generalities

Before dosimetric calculations from ^177^Lu-PSMA images, validating prior control phases is crucial to manage the uncertainty in dose estimation. This ensures reliable personalized activity administration for future treatments. Numerous sources of variability affect this process, and current commercial solutions lack uncertainty values, making it essential for physicists to establish robust methodologies to minimize errors and ensure accurate dosimetry [7].

#### 2.2.2. Calibration of Activimeters

Activimeters must be calibrated using a traceable source from a primary standards laboratory for each container (syringe and vial) used to measure ^177^Lu. This ensures the reliability of the uncertainty associated with activity measurements. Furthermore, the stability of the activity meter’s response must be monitored for each channel used to detect any potential drift. Using a traceable source ensures that the measurements are accurate and reliable, which is crucial for precise dose administration, optimizing treatment efficacy, and minimizing risks. Each type of container, whether syringes or vials, must be calibrated individually to maintain accurate measurements. Monitoring the stability of the activity meter’s response involves regularly checking measurements against known standards to detect any drift. If drift is detected, recalibration or adjustments must be made to ensure measurement accuracy. These practices are essential to guarantee that the doses administered to patients are accurate, maximizing therapeutic effectiveness while minimizing the risks associated with incorrect dose administration [2].

#### 2.2.3. Image Quantification

Estimating the final dose requires accurate image quantification, addressing variability sources such as attenuation, scatter and spatial resolution corrections, dead time correction, partial volume effect (PVE) correction, and calibration factor determination. These factors convert image counts to activity in Becquerels. Each correction step enhances the accuracy of the dosimetric calculations by ensuring that the quantified images reflect true activity levels within the body.

#### 2.2.4. Calibration Factor

The calibration factor, determined using a cylindrical phantom with a known radioactive concentration, converts SPECT data counts to activity. This process accounts for patient heterogeneity by performing multiple acquisitions over varying activities and using linear regression to calculate the calibration coefficient. Accurate calibration factors are essential for converting image data into reliable activity measurements, directly impacting the dosimetric calculations [8].

#### 2.2.5. Partial Volume Effect Correction

PVE impacts lesion detection and uptake intensity, with underestimation ranging from 60% to over 99%. Correction methods include system modeling during reconstruction or post-reconstruction applications, such as the recovery coefficient (RC) method and extended volume of interest (VOI) method. Accurate PVE correction is crucial for reliable dosimetry, as it ensures that the true intensity of radioactive uptake within lesions and organs is measured correctly, leading to more precise dose estimates [9,10,11,12,13]; Figure 2.

#### 2.2.6. Image Acquisition and Reconstruction for Dosimetry

To assess ^177^Lu-PSMA uptake dynamics, multiple time-point acquisitions are needed, ideally using SPECT-CT for each point to avoid superimposition errors. Medium-energy collimators and optimized reconstruction parameters enhance image quality. Scattering and attenuation corrections are mandatory for quantitative accuracy, typically involving dual or triple energy window methods. These steps ensure that the images accurately represent the distribution of radioactivity within the body, essential for calculating absorbed doses [2,14,15,16].

#### 2.2.7. Volume Delineation

Quantifying activity in a region requires measuring counts in the corresponding VOI, with images aligned to a reference scan. VOIs can be segmented on the reference scan or at each time point, depending on structure volume changes. Automatic delineation tools reduce variability and improve reproducibility. Accurate volume delineation is critical for dosimetry, as it defines the regions where activity measurements are taken, directly affecting the calculation of absorbed doses [17,18,19,20].

#### 2.2.8. Pharmacokinetic Curves

Pharmacokinetic curves for ^177^Lu-PSMA illustrate activity changes over time within an organ or tumor. Constructed using SPECT-CT data points, these curves help determine cumulative activity. Accurate curves depend on multiple data points and appropriate analytical models. These curves are essential for understanding how the radiopharmaceutical behaves in the body over time, providing the basis for calculating the total absorbed dose in various tissues [21,22,23].

#### 2.2.9. Calculation of the Absorbed Dose

Absorbed dose calculations in internal radiotherapy rely on the medical internal radiation dose (MIRD) methodology. The S factor, representing dose absorbed per unit of cumulative activity, is determined using Monte Carlo simulations. Personalized dosimetry involves software like Voxel Dosimetry™, PLANET^®^ Onco Dose, and SurePlan^®^ MRT, which calculate doses at the voxel level using patient-specific data and Monte Carlo or dose voxel kernel (DVK) convolution methods. These software tools integrate anatomical and metabolic data to provide precise dosimetric calculations, ensuring that the absorbed doses are accurately determined for each patient.

### 2.3. Dosimetric Data by Organ and Tumor

To address a table for that, the methodology consists of a search on two databases: PubMed (Medline) and Science Direct, with the key words PSMA, RLT, and dosimetry. We have excluded the case reports and reviews. There were 29 results finally included. The flowchart is shown in Figure 3.

The organs at risk (OAR) in ^177^Lu-PSMA-RLT are essentially salivary glands, kidneys, and bone marrow [24]. Considering the lachrymal glands as OAR in clinical routine is controversial [25,26]. Additionally, while there is notable uptake in the liver, spleen, and digestive tract, these organs are generally not classified as OAR [27].

Among the 29 dosimetry studies examined, most of them (22) provided absorbed dose estimates for both kidneys and salivary glands, nearly two third of them investigated the bone marrow (19), and nearly one third of them investigated the lacrimal glands. The median absorbed dose per unit of GBq activity for each organ was as follows:For kidneys: 0.55 Gy/GBq ± 0.27For salivary glands: 0.81 Gy/GBq ± 0.19For the lacrimal glands: 2.26 Gy/GBq ± 1.02For bone marrow: 0.03 Gy/GBq ± 0.007

The data on absorbed doses in healthy organs during PSMA therapy exhibit considerable heterogeneity. For example, with small molecules like ^177^Lu-PSMA-617, there have been significant variations in calculated marrow doses.

The absorbed doses in tumors vary depending on the site of metastasis: bone metastases (26 ± 20 Gy), lymph node metastases (24 ± 16 Gy), and pulmonary metastases (13 ± 7.4 Gy). The average effective dose for tumors is 0.08 ± 0.07 mSv/MBq. In cases of repeated treatments, the absorbed doses in various OAR tend to remain relatively stable, but the delivered dose to the tumor tends to decrease with each treatment cycle.

Table 1 provides a summary of the mean absorbed doses (in mGy/MBq) for both the tumor and the key OAR. The results were displayed with mean ± (standard deviation) SD (if available).

### 2.4. Various Dosimetric Imaging Techniques

A number of different dosimetric approaches have been suggested or employed, and it seems that they can be broadly categorized into four groups: 2D methods, 3D methods, combined methods, and hybrid methods.

2D Method: In the 2D method, a personalized dosimetry plan is derived from whole-body planar scintigraphy, after which software tools, often OLINDA/EXM, are used to compute absorbed organ doses. In particular, OARs, such as the kidneys, and tumors can be identified, and their uptake is used to calculate the absorbed dose. This approach was exclusively used by Yadav et al., Scarpa et al., and Okamoto et al. [32,35,36]. It is worth noting that Yadav et al. conducted measurements at nine time points. We would like to propose the following time points for further consideration: 0.5, 3.5, 24, 48, 72, 96, 120, 144, and 168 h after RTL administrations. In their approach, Scarpa et al. proposed timepoints at 0.5, 4, 24, 72, and 96 h, while Okamoto et al. suggested timepoints at 0.5, 2, 24, and 144 h after administration, employing PSMA-617 and PSMA-I&T radioligands [35].

With regard to the 3D method, we would like to propose the following: The 3D method involves a conversion of voxel values obtained from SPECT/CT scans into activity values using calibration factors and a correction with volume-dependent ^177^Lu recovery coefficients. Violet et al. exclusively used this method, conducting quantitative SPECT/CT scans (2- or 3-bed-position acquisition) covering the neck to pelvis at 4-, 24-, and 96-h post-administration [40].

Perhaps it would be beneficial to consider combining methods. I would be remiss if I did not mention that Delker et al. and Kabasakal et al. employed combined methods, which involved using both 2D and 3D approaches. In their study, Delker et al. conducted whole-body planar and SPECT/CT imaging of the abdomen at 1-, 24-, 48-, and 72-h post-administration [30] Kabasakal et al. conducted whole-body scans at 4-, 24-, 48-, and 120-h post-administration, and only performed SPECT/CT imaging after 24 h [34].

Hybrid Method: Rosar et al. proposed a hybrid dosimetry method that combines three planar whole-body scintigraphy with one SPECT/CT scan for calibration of the respective time–activity curve. This method seeks to integrate the 2D kinetics derived from planar scintigraphy within source regions with the activity estimation from the SPECT scan conducted on day 1 or day 2. In their recent study, Rosar et al. evaluated a total of 65 cycles of ^177^Lu-PSMA-617 RLT in 24 patients. They compared these three methods, with the hybrid method proving to be the most promising, offering the potential to replace the 3D method [48].

Similarly, Kurth et al. introduced two hybrid dosimetry methods and compared them to a “full dosimetric scheme”. This study, which involved 46 patients, employed the “full dosimetric scheme”, which included whole-body scintigraphy and two-bed SPECT/CT imaging at all four time points (approximately 2, 24, 48, and 72 h after administration) [50].

The different dosimetry methods for post-therapeutic dosimetry in ^177^Lu-PSMA-617 radioligand therapy each have their distinct advantages.

2D methods are praised for their time efficiency, simplicity, and cost-effectiveness. These methods are less time-consuming compared to 3D methods, making them practical for routine clinical use as planar whole-body scintigraphy requires approximately 15 min per scan. They are straightforward and do not necessitate the complex equipment and software needed for 3D dosimetry. Additionally, due to their simpler technology and shorter scan times, 2D methods are generally more cost-effective.

On the other hand, 3D methods are known for their high accuracy and detailed analysis capabilities. They provide the most accurate dose estimates by involving detailed volumetric imaging and precise activity quantification, which avoids the underestimation issues often seen in 2D methods. The ability to perform detailed spatial analysis of dose distribution within the body allows for better targeting of tumors and sparing of healthy tissues. Moreover, 3D dosimetry benefits from reduced inter-observer variation through automatic volume delineation and advanced image reconstruction techniques, leading to more consistent results.

Hybrid methods offer a balance between accuracy and efficiency. By combining the speed and simplicity of 2D methods with the accuracy of 3D methods, hybrid dosimetry uses a single SPECT/CT scan to calibrate planar images. This approach provides dose estimates nearly equivalent to those from 3D methods while being feasible for routine clinical use due to reduced scan times compared to full 3D dosimetry. Additionally, the shorter scan times required for hybrid dosimetry increase patient compliance and comfort, making it a more practical option for repeated measurements.

Table 2 briefly provides a summary of the different methods of dosimetry.

## 3. Discussion on the Post-Therapeutic Imaging

### 3.1. EANM Recommendations on Personalized Dosimetry

EANM recommendations, which are based on the European directive 2013/59/Euratom, suggest that it may be beneficial to plan and verify exposures of target volumes individually, while aiming to minimize doses to non-target volumes. It is worth noting that established tolerance limits for various OAR include 2 Gy for red marrow (single exposure), 28–40 Gy for kidneys (depending on risk factors), and 35 Gy for salivary glands. It is important to note that these recommendations highlight the potential for impaired renal function to significantly elevate absorbed doses in organs, particularly red marrow, due to PSMA-expressing bone lesions.

In light of the difficulties posed by multiple post-therapeutic imaging sessions, the recommendations put forward a proposal to consider simplified methodologies. It is recommended that the minimum standard involves dosimetry based on a single imaging time point, preferably using quantitative 3D techniques, conducted at least three or more days post-application.

However, another position paper categorizes ^177^Lu RLT, as well as ^177^Lu-DOTATATE and ^177^Lu-PSMA-ligands, as level 1, which implies activity-based prescription and patient-averaged dosimetry [3].

### 3.2. Alternative Practices without Imaging Based Personalized Dosimetry

The VISION study, a large multi-center phase 3 comparative study, administered a fixed dose of 7.4 GBq (200 mCi) of ^177^Lu-PSMA-617 every six weeks for four cycles, without incorporating personalized dosimetry considerations. In cases where patients showed a response, an additional two cycles could be administered, for a total of up to six cycles. The study did not uniformly include post-therapeutic imaging, reflecting variations in practices across multiple centers.

This trend towards “non-personalized dosimetry” is becoming more common, raising questions about the necessity of personalized dosimetry as recommended by the European Association of Nuclear Medicine. Unlike I-131 post-therapeutic whole-body imaging, which assesses disease extension, ^177^Lu-PSMA-RLT primarily relies on Ga-68 PSMA PET imaging and biological parameters for precision and prediction.

Despite the shift towards fixed dosing, initial data suggest a dose-response relationship in PSMA RLT, potentially indicating a role for dosimetry at the lesion level. For instance, Violet et al. observed a correlation between whole-body tumor-absorbed doses and PSA decline, an encouraging finding.

### 3.3. Controversy

#### 3.3.1. Arguments for Dosimetry Based on the Post-Therapeutic Imaging

Improved treatment efficacy and safety can be achieved through post-therapeutic imaging, which allows for personalized dosimetry and the precise calculation of the radiation dose tailored to each patient. This optimization enhances therapeutic effects while minimizing damage to healthy tissues, leading to better treatment outcomes and fewer side effects compared to standardized dosing methods. Personalized dosimetry significantly reduces the risk of treatment-related toxicity by ensuring that critical organs receive doses within safe limits, thereby preventing severe side effects from overexposure. Additionally, it provides detailed information on how different radiation doses affect patient outcomes, refining treatment protocols and enhancing the understanding of dose-response relationships crucial for developing more effective therapies. For patients with complex medical histories or multiple comorbidities, personalized dosimetry allows for careful planning of radiation doses to avoid exacerbating existing conditions or introducing new complications, thus better managing these complex cases [51,52,53].

#### 3.3.2. Arguments against Post-Therapeutic Imaging

Post-treatment dosimetry does not change the treatment. At best, post-treatment dosimetry can provide information about the likelihood of success or the likelihood of side effects. Moreover, the clinical benefit of dosimetry in RLT is not yet well demonstrated. Although there are dose-effect correlations, such as between bone marrow absorbed dose and platelets in various treatments, these correlations have not translated into clear improvements in treatment outcomes like PFS or OS [51].

Implementing dosimetry involves significant economic and logistical challenges, including the need for additional equipment, such as new imaging scanners, and increased staffing for technologists, physicians, and medical physicists to handle the increased workload. The complexity and cost of these requirements may outweigh the benefits, especially in the absence of definitive evidence of improved clinical outcomes. There is also a global shortage of healthcare professionals in nuclear medicine, including nuclear medicine physicians, radiopharmacists, medical physicists, and technologists. Given the increasing demand for RLT and stable healthcare costs, resources should be focused on clinically relevant tasks with proven benefits [2].

Modifying the dosage of approved drugs based on dosimetry outside clinical trials is considered off-label use. This practice is highly supervised and involves personal responsibility for any adverse events. Physicians must inform patients and obtain their consent for off-label use, making it a less favorable option without strong supporting evidence of safety and efficacy [51].

Dosimetry often requires multiple imaging sessions over several days, which can be burdensome for patients, particularly those who need to travel long distances or have limited social support. This increased burden may lead to reluctance in patient compliance, especially if the benefits of dosimetry are not clearly demonstrated.

There is also significant heterogeneity in how dosimetry is performed across institutions. Without standardized protocols and harmonization of software, lesion segmentation, and scanner qualification, it is challenging to implement dosimetry uniformly in clinical practice. This lack of standardization can lead to inconsistent results and limited applicability [2].

For certain aspects of treatment, such as monitoring bone marrow toxicity, alternative techniques like blood cell counts may be more economically and clinically efficient than dosimetry. These techniques are already established and may suffice in monitoring specific toxicities without the need for complex dosimetric calculations [54,55].

### 3.4. Role of Pretherapeutic Imaging in Personalized Dosimetry

Simplifying dosimetric approaches involves tapping into valuable information from pretherapeutic imaging. Given the stability of ^68^Ga PSMA PET-CT in ^177^Lu-PSMA-RLT, particularly in staging, it holds significant potential.

However, only a limited number of studies have delved into this area. Violet et al. established meaningful correlations between pretherapeutic ^68^Ga-PSMA PET and estimated doses for tumors, salivary glands, and bone marrow. They also noted an inverse correlation between tumor volume (defined on PSMA PET) and mean doses to parotid glands and kidneys. Furthermore, they discovered a significant correlation between the SUVmean of the “whole-body” tumor on screening ^68^Ga-PSMA PET and the “whole-body dose” (Gy/GBq). A similar correlation was suggested by Scapa et al. using SUVmax or SUVmean from pre-therapeutic PSMA-11 PET/CT to predict delivered doses to tumors [40].

Scarpa et al. and Okamoto et al. reported only moderate correlations between pretherapeutic PSMA PET/CT SUVs and absorbed dose estimates [35,36]. Other teams have also investigated the correlation between PSMA PET/CT data and radiation absorbed doses (ADs) following ^177^Lu-PSMA therapy. Their aim was to determine if pretherapeutic PSMA PET/CT studies could predict AD and thereby ensure treatment safety and efficacy. Hohberg et al. observed significant differences (*p* < 0.01) in the ratios of lymph node, bone lesion, and tumor uptake to kidney uptake on pretherapeutic PSMA PET/CT between responders and non-responders (with response defined as a PSA level decline of at least 50%). Furthermore, the ratio of tumor-to-kidney uptake correlated with the mean radiation AD in both responders and non-responders. The authors suggest that such correlations could enable the estimation of therapeutic response from pretherapeutic PSMA PET/CT scans and the identification of potential responders before the first cycle of ^177^Lu-RLT therapy [56].

### 3.5. Summary

While dosimetry offers potential benefits for optimizing radionuclide therapy, several significant challenges must be addressed. These include demonstrating clinical benefits in radionuclide therapy, overcoming economic and logistical hurdles, managing resource allocation, navigating regulatory frameworks, minimizing patient burden, and addressing heterogeneity in dosimetry practices. Addressing these challenges requires further research, standardized protocols, and a multidisciplinary approach to effectively integrate dosimetry into routine clinical practice.

Firstly, incorporating dosimetry into clinical trials more frequently is necessary to show correlations, if they exist, with relevant clinical outcomes such as PFS or OS. Improving the practice of post-treatment dosimetry through streamlined single SPECT/CT scan schemes is also essential, as the total integrated activity and radiation dose can be estimated from a single time sample. Ideally, dosimetry based on post-therapeutic imaging should evolve into dosimetry based on pretherapeutic imaging.

The role of dosimetry in radionuclide therapy has not yet been fully established. Although various institutional protocols exist, current guidelines suggest that complete dosimetry is optional for standard treatments. However, given significant patient variability, patient-specific dosimetry may be beneficial, especially for treatments administered early in the disease course or across multiple treatment cycles.

In conclusion, a balanced approach is prudent to determine the relevance and benefits of a full scheme of post-therapeutic imaging for personalized dosimetry. This approach should consider patient-specific factors and evolving research in the field, aiming to optimize treatment efficacy and minimize side effects.

## 4. Conclusions

The ^177^Lu-labelled RLT targeting PSMA offers promising treatment options for patients with progressive metastasized mCRPC. Personalized dosimetry based on post-therapeutic imaging holds significant potential to improve treatment outcomes by tailoring radiation doses to individual patient needs, thereby maximizing therapeutic efficacy while minimizing toxicity to healthy tissues. However, the implementation of personalized dosimetry is challenging due to the resource-intensive nature of the required multiple SPECT/CT scans and the logistical burdens it places on healthcare facilities and patients.

The benefits of personalized dosimetry include optimized radiation delivery, enhanced safety, and better management of complex cases. Conversely, the drawbacks involve increased patient burden, significant resource constraints, and implementation challenges. While the EANM advocates for personalized dosimetry, the practice is not yet standardized due to these practical challenges. Further research and streamlined methodologies are needed to balance the advantages of personalized dosimetry with its feasibility in routine clinical practice.

Overall, the integration of personalized dosimetry into clinical protocols for ^177^Lu-PSMA-RLT could significantly enhance treatment outcomes for mCRPC patients, provided that the associated logistical and resource challenges can be effectively managed.

## Figures and Tables

**Figure 1 biomedicines-12-01512-f001:**
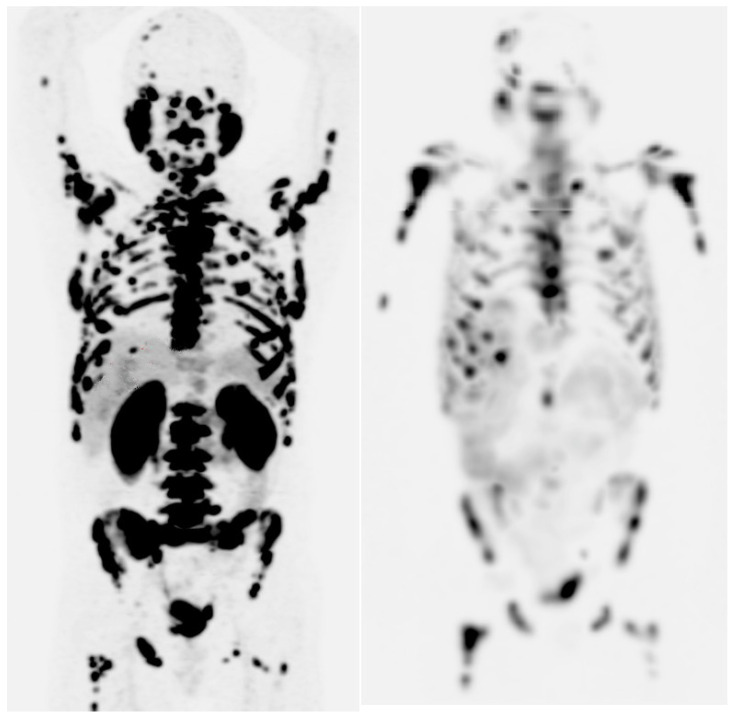
Maximum intensity projection images (MIP) of initial ^68^Ga-PSMA-11 PET (left one) and MIP images of post-treatment SPECT performed 24 h after the first cycle of radioligand therapy with ^177^Lu PSMA-617. There is a good correlation of different metastasis sites.

**Figure 2 biomedicines-12-01512-f002:**
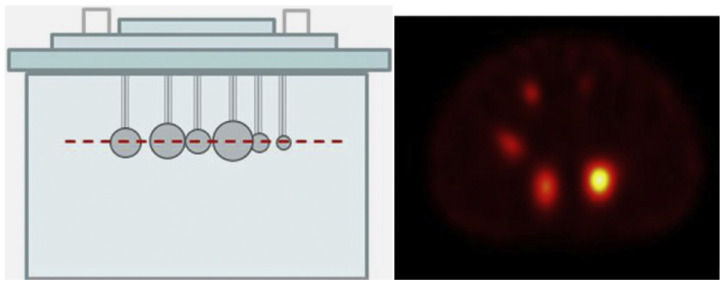
Phantom and cross section of a reconstructed image to visualize the partial volume effect on spheres of different sizes containing the same radioactive concentration. On the left-hand side, the diagram, and on the right-hand side, the scan image.

**Figure 3 biomedicines-12-01512-f003:**
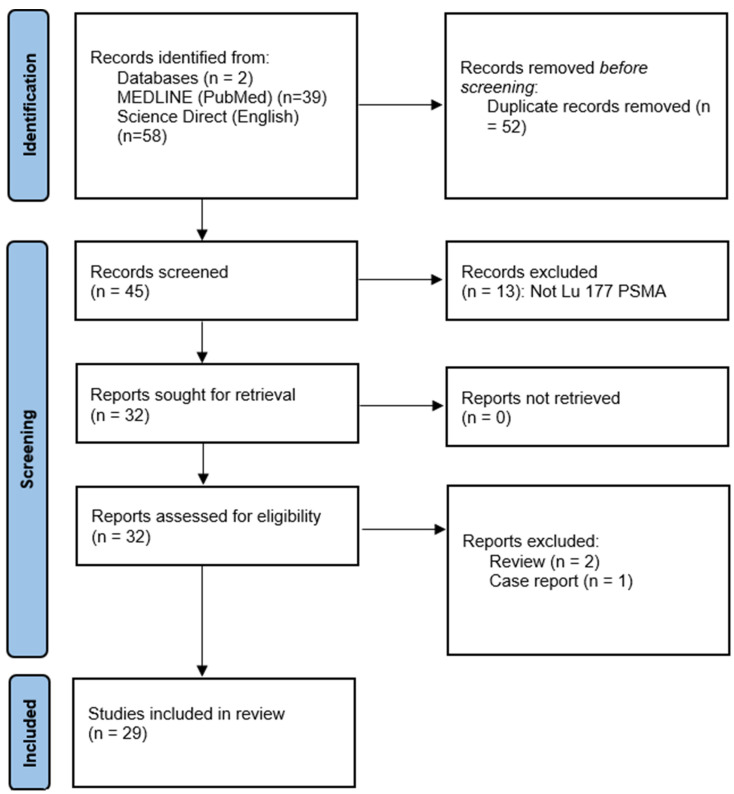
Flowchart of the selection of dosimetry studies.

**Table 1 biomedicines-12-01512-t001:** Absorbed doses for tumors and key organs of risk.

Author Year Reference	Ligand	Tumor	Kidneys	Parotid Gland	Lacrimal Gland	Bone Marrow
Kabasakal 2015 [22]	PSMA-617	NS	0.88 ± 0.40	1.17 ± 0.31	NS	0.03 ± 0.01
Baum 2016 [28]	PSMA-I&T	NS	0.8	1.3	NS	NS
Delker 2016 [29]	PSMA-617	13.1	0.6	1.4	NS	0.012 ± 0.005
Hohberg 2016 [30]	PSMA-617	NS	0.53 ± 0.17	0.72 ± 0.14	2.82 ± 0.76	NS
Kratochwil 2016 [31]	PSMA-617	12.05 ± 7.07	0.75 ± 0.19	1.28 ± 0.40	NS	0.03 ± 0.01
Yadav 2016 [32]	PSMA-617	5.28 ± 2.46	0.99 ± 0.31	1.24 ± 0.26	NS	0.048 ± 0.05
Fendler 2017 [33]	PSMA-617	6.1 ± 4.9	0.55 ± 0.2	1.0 ± 0.6	NS	0.002 ± 0.005
Kabasakal 2017 [34]	PSMA-617	NS	0.82 ± 0.25	1.90 ± 1.19	NS	0.030 ± 0.008
Okamoto 2017 [35]	PSMA-I&T	4.2 ± 5.3	0.72 ± 0.21	0.55 ± 0.14	3.8 ± 1.4	NS
Scarpa 2017 [36]	PSMA-617	2.1 ± 0.8	0.60 ± 0.36	0.56 ± 0.25	1.01 ± 0.69	0.04 ± 0.03
Gosewisch 2018 [37]	PSMA-617	NS	NS	NS	NS	0.0108
Gosewisch 2019 [38]	PSMA-617	NS	NS	NS	NS	0.012
Sarnelli 2019 [39]	PSMA-617	NS	0.67 ± 0.27	0.81 ± 0.74	NS	0.044 ± 0.017
Violet 2019 [40]	PSMA-617	15.71 ± 14.72	0.39 ± 0.15	0.58 ± 0.43	0.36 ± 0.18	0.11 ± 0.10
Ozkan 2020 [41]	PSMA-617	NS	0.70 ± 0.24	1.34 ± 0.78	2.28 ± 1.29	NS
Paganelli 2020 [42]	PSMA-617	4.70	0.42	0.65	2.26	0.036
Chatachot 2021 [43]	PSMA-I&T	NS	0.81 ± 0.24	0.21 ± 0.14	3.62 ± 1.78	0.02 ± 0.01
Feuerecker 2021 [44]	PSMA-I&T	2.64 ± 2.24	0.73 ± 0.18	0.80 ± 0.41	NS	0.30 ± 0.27
Kamaldeep 2021 [23]	PSMA-617	9.92 ± 3.02	0.49 ± 0.17	0.53 ± 0.25	1.23 ± 0.70	0.03 ± 0.02
Mahmoudi 2021 [45]	PSMA-617	NS	0.46 ± 0.09	0.62 ± 0.07	NS	NS
Peters 2021 [46]	PSMA-617	3.25 ± 3.19	0.49 ± 0.11	0.39 ± 0.17	NS	0.017 ± 0.008
Prive 2021 [47]	PSMA-617	2.14 ± 1.83	0.49 ± 0.11	0.39 ± 0.17	NS	0.02 ± 0.01
Rosar 2021 [48]	PSMA-617	1.42 ± 0.99 (2D) 1.68 ± 1.32 (3D) 1.55 ± 1.28 (hybrid)	0.49 ± 0.31 (2D) 0.54 ± 0.28 (3D) 0.52 ± 0.27 (hybrid)	0.75 ± 0.34 (2D) 0.81 ± 0.34 (3D) 0.81 ± 0.34 (hybrid)	NS	NS
Schuchardt 2021 [49]	PSMA-617 and PSMA-I&T	5.8 for PSMA-617 and 5.9 for PSMA-I&T	0.77 for PSMA-617 and 0.92 for PSMA-I&T	0.5 for PSMA-617 and PSMA-I&T	5.1 for PSMA-617 and 3.7 for PSMA-I&T	NS
Volter 2021 [50]	PSMA-617	Skeletal: 4.7 ± 3.9, Nodal: 7.7 ± 9.7	NS	NS	NS	NS
Mix 2022 [51]	PSMA-617	NS	0.67 ± 0.24	NS	NS	NS
Uijen 2023 [52]	PSMA-617 and PSMA-I&T	NS	0.49 for PSMA-617 and 0.73 for PSMA-I&T	NS	NS	NS

NS: not stated. PSMA: prostate-specific membrane antigen.

**Table 2 biomedicines-12-01512-t002:** Different published methods of dosimetry for ^177^Lu-PSMA radioligand therapy.

Author Year Reference	Imaging Protocol
Delker 2016 [29]	Serial Planar and SPECT
Fendler 2017 [33]	Serial Planar and SPECT
Gosewisch 2018	3D & Planar w/blood sampling
[37]	
Gosewisch 2019	Serial SPECT w/single planar
[38]	
Hohberg 2016 [30]	Serial Planar
Kabasakal 2015 [22]	Serial Planar w/bloodsampling
Kratochwil 2016 [31]	Serial Planar + single SPECT w/blood sampling
Mix 2021 [51] Paganelli 2020 [42]	Serial SPECT Serial Planar + single SPECT
Peters 2021 [46]	Serial SPECT
Prive 2021 [47]	Serial SPECT w/blood sampling
Rosar 2021 [48]	Serial Planar and SPECT
Sarnelli 2019 [39]	Serial Planar w/blood sampling
Scarpa 2017 [36]	Serial Planar + single SPECT
Violet 2019 [40]	Serial SPECT w/blood
Volter 2021 [50]	Serial SPECT sampling
Yadav 2016 [32]	Serial Planar w/blood

SPECT: single-photon emission-computed tomography.

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
