# Peer review of "Lutetium-177-Prostate-Specific Membrane Antigen Radioligand Therapy: What Is the Value of Post-Therapeutic Imaging?"

_biomedicines, 2024, doi:10.3390/biomedicines12071512_

Round 1

Reviewer 1 Report

Comments and Suggestions for Authors

The manuscript by Jules Zhang-Yin submitted to Biomedicines intends to explore the role of post-therapeutic imaging in patients undergoing Lutetium-177 (Lu-177) labelled prostate-specific membrane antigen (PSMA) radioligand therapies (Lu-177-PSMA-RLT). The manuscript is well written and interesting. The interpretations and conclusions are justified. The manuscript seems to be of broad interest for chemist and biologist working in the area. Therefore, it is recommended for publication with the following issues being addressed appropriately.

Some revisions are as follows:

1.      Throughout the whole manuscript, 177Lu should be corrected as 177Lu.

2.      Page 2, line 56, because “metastatic castration-resistant prostate cancer (mCRPC)” has been shown in the abstract section, “metastatic castration-resistant prostate cancer (mCRPC)” should be corrected as mCRPC.

3.      177Lu-PSMA include several different 177Lu-labelled complexes, so the authors should afford the exact structures and names of 177Lu-PSMA.

4.      In particular, there are several format issues in References section.

Author Response

I would like to thank you sincerely for your constructive criticism and valuable comments:

My response follows: 

  1. Throughout the whole manuscript, 177Lu should be corrected as177

R: the change has been made.

  1. Page 2, line 56, because “metastatic castration-resistant prostate cancer (mCRPC)” has been shown in the abstract section, “metastatic castration-resistant prostate cancer (mCRPC)” should be corrected as mCRPC.

R: the change has been made.

  1. 177Lu-PSMA include several different 177Lu-labelled complexes, so the authors should afford the exact structures and names of 177Lu-PSMA.

R: the specifications have been added.

  1. In particular, there are several format issues in References section.

R: The references have been revised to standardise the format

Reviewer 2 Report

Comments and Suggestions for Authors

1. This is a confusing review on an important emerging topic. Ligand based radiotherapy presents some unique challenges. We can expect that more agents in this class are coming; the lesson we learn from Lu-PSMA will be applied to them. The problem with this review is that it is hard to understand what is trying to be. There is already a white paper on this topic jointly authored by EANM and SNMMI (Kratochwil, C. et al.  Eur J Nucl Med Mol Imaging 50, 2830–2845 (2023).) The review also does not clearly address it’s stated questions. At the end of the introduction we are told that the focus will be on personalized dosimetry but we don’t really get into the dosimetry discussion until line 380. I think there is something of value here but the author really needs to focus on what they actually want to review and discuss. My broad comments are below: 

1. On the title: “Which Place for Post Therapeutic Imaging?” This is not grammatically correct, particularly the phrasing of “which place”. Is the focus on when should imaging be performed? Is the focus on the value of post therapeutic imaging? Neither of these are places.

2. Line 24: The manuscript starts by talking about the Vision study which is not introduced properly until line 59.

3. Section 2.1 on the procedures is too long and mostly unnecessary if the focus is on the dosimetry. We need some information on procedures particularly as it relates to the dose delivered as described in section 2.1.3. I will note that this section is missing the required pre-scan with a PSMA PET or SPECT agent which does get discussed later in section 3.3. This therapy would not be given if the tumor does not express PSMA after all

4. Section 2.2 on how to do dosimetry is too long and can be summarized. If there are points here that affects some of the considerations latter then that is when they should be brought up. In combination with the previous point, the problem, here is that the introduction is too long and the discussion of the challenges is too short.

5. Line 381-383. Please look up the PRISMA guidelines for the proper way of performing and reporting this search.

6. The sentence on lines 404-405 should be on line 383 to finish up the first paragraph of this section. Otherwise, you are discussing the results presented in the table before you mention where the data is being presented.

7. The median dose should include some uncertainty and be included in Table 1

8. Table 2 could be combined with Table 1

9. Section 2.4 seems to focus on two different and important points 2D vs 3D and timeline of scanning. Neither is discussed enough. We are told how it was done in a certain paper or two and then how the author would like to se it done. But no reason is given for the author’s suggestions. There is also no discussion of the advantages and disadvantages of each method.

10. Section 3 should start with the recommendations from EANM/SNMMI. 

11. Very little spaces is given to the controversies presented in section 3.1. I would also point out that I would have one argument against pos-treatment dosimetry that I do not see presented: Post-treatment dosimetry does not change the treatment. At best post-treatment dosimetry can inform on the likelihood of success or likelihood of side effects. 

12. I think section 3.3 can be expanded to discuss the potential of patient-specific dosimetry based on pre-treatment imaging particularly as it relates to the possibility of patient-specific dosing.

13. Section 3.4 presents the problem discussed rather well and it may be valuable to have this earlier in the manuscript. It would serve as a good introduction to the problem

14. The paragraph in lines 547-550 is incomplete and is meaningless fluff that should be removed.

Author Response

Dear reviewer, I would like to sincerely thank you for your constructive criticism and valuable comments, which were a great help in revising the paper. I apologise for the confusing aspect of the first version and hope it meets your standards. I'm available to improve it further if necessary. 

My response follows:

  1. This is a confusing review on an important emerging topic. Ligand based radiotherapy presents some unique challenges. We can expect that more agents in this class are coming; the lesson we learn from Lu-PSMA will be applied to them. The problem with this review is that it is hard to understand what is trying to be. There is already a white paper on this topic jointly authored by EANM and SNMMI (Kratochwil, C. et al.  Eur J Nucl Med Mol Imaging 50, 2830–2845 (2023).) The review also does not clearly address it’s stated questions. At the end of the introduction we are told that the focus will be on personalized dosimetry but we don’t really get into the dosimetry discussion until line 380. I think there is something of value here but the author really needs to focus on what they actually want to review and discuss. My broad comments are below: 
  2. On the title: “Which Place for Post Therapeutic Imaging?” This is not grammatically correct, particularly the phrasing of “which place”. Is the focus on when should imaging be performed? Is the focus on the value of post therapeutic imaging? Neither of these are places.

R: Sorry for this confusing title. I would like to discuss the value of post therapeutic imaging.

I propose the modified title: “177Lu-PSMA Radioligand Therapy: what is the value of post-therapeutic imaging ?”

  1. Line 24: The manuscript starts by talking about the Vision study which is not introduced properly until line 59.

R:  The change has been made in line 29. It was made with the correct introduction of the Vision study for the first time.

  1. Section 2.1 on the procedures is too long and mostly unnecessary if the focus is on the dosimetry. We need some information on procedures particularly as it relates to the dose delivered as described in section 2.1.3. I will note that this section is missing the required pre-scan with a PSMA PET or SPECT agent which does get discussed later in section 3.3.

R: Thank you indeed for that comment. Section 2.1 has been significantly shortened. And I have added, as you suggested, the paragraph on the required pre-scan with a PSMA PET or SPECT agent.(Line 91)

  1. Section 2.2 on how to do dosimetry is too long and can be summarized. If there are points here that affects some of the considerations latter then that is when they should be brought up. In combination with the previous point, the problem, here is that the introduction is too long and the discussion of the challenges is too short.

R: This section has been significantly shortened and summarized.

  1. Line 381-383. Please look up the PRISMA guidelines for the proper way of performing and reporting this search.

R: It has been modified in line 190 and a flowchart has been added.

  1. The sentence on lines 404-405 should be on line 383 to finish up the first paragraph of this section. Otherwise, you are discussing the results presented in the table before you mention where the data is being presented.

R: The change has been made by moving the sentence to the correct place.

  1. The median dose should include some uncertainty and be included in Table 1

R: The change has been made by adding the SD and included in the Table 1.

  1. Table 2 could be combined with Table 1

R: I'm sorry that I can't do this, as there are too many elements when they're put together.

  1. Section 2.4 seems to focus on two different and important points 2D vs 3D and timeline of scanning. Neither is discussed enough. We are told how it was done in a certain paper or two and then how the author would like to se it done. But no reason is given for the author’s suggestions. There is also no discussion of the advantages and disadvantages of each method.

R: This section has been significantly broadened.

  1. Section 3 should start with the recommendations from EANM/SNMMI. 

R: The change has been made.

  1. Very little spaces is given to the controversies presented in section 3.1. I would also point out that I would have one argument against pos-treatment dosimetry that I do not see presented: Post-treatment dosimetry does not change the treatment. At best post-treatment dosimetry can inform on the likelihood of success or likelihood of side effects. 

R: This section has been significantly broadened.

  1. I think section 3.3 can be expanded to discuss the potential of patient-specific dosimetry based on pre-treatment imaging particularly as it relates to the possibility of patient-specific dosing.

R: Surely this section has been developed and expanded.

  1. Section 3.4 presents the problem discussed rather well and it may be valuable to have this earlier in the manuscript. It would serve as a good introduction to the problem

R: This section has been moved to the earlier section, just after the EANM recommendations.

  1. The paragraph in lines 547-550 is incomplete and is meaningless fluff that should be removed.

R: It has been removed.

Reviewer 3 Report

Comments and Suggestions for Authors

Authors presented a potential good strategy for prostate cancer patient, however the writing of this manuscript may need major revision to improve the current state : 

1.        Abstract – define SPECT

2.        Introduction – a short description of “VISION Study” may help others to understand the context better.

3.        No reference was cited in most of the statements in Section 2.1, 2.2 Authors should include all the necessary references. Are they from single source? Company’s insert? Is it potential bias? Any conflict of interests by authors when drafting this manuscript should be declared.

4.        Section 2.1 should include more evidence of how other study has observed similar observation. This section was written likes a product insert which could be lack of scientific and critical review/inputs from literature.

5.        Several flow charts or figures in relevant sections may help the readers to understand better.

6.        Line – 167 “Additionally, monitoring the stability of the activimeters' response must be carried out for each channel used to detect any potential drift.” – Kindly elaborate how this should be done?

7.        Section 2.2 should be repositioned to Section 3, a stand-alone section.

8.        The “+/-“ should be replaced with symbol

9.        Missing Title for table 1 and table 2.

10.  Abbreviation used in the table should be defined in footnote.

11.  Section 3.1.1 and 3.1.2 should be rewritten in paragraph format rather than “XXXX :” Alternative these point forms should be converted to graphic.

12.  The discussion may need more critical rewrite in such whats the take home message for the readers? whats the gap of knowledge? whats authors suggestion for next actions for the way forward.

13. Abbreviation should be defined at first use, example but not limited to:

a.        Line 65 – define EANM/SNMMI

b.        Line 68 – define ANSM

c.        Line 90 – define RLT

Comments on the Quality of English Language

Minor editing is required for typo-errors and abbreviation should be defined at first use. 

Author Response

Dear reviewer, I would like to sincerely thank you for your constructive criticism and valuable comments, which were a great help in revising the paper. 

My response follows:

  1. Abstract – define SPECT

R: The change has been made.

  1. Introduction – a short description of “VISION Study” may help others to understand the context better.

R: The change has been made in line 24

  1. No reference was cited in most of the statements in Section 2.1, 2.2 Authors should include all the necessary references. Are they from single source? Company’s insert? Is it potential bias? Any conflict of interests by authors when drafting this manuscript should be declared.

R: These two sections have been thoroughly revised and summarized. I hope it meets your standards.

  1. Section 2.1 should include more evidence of how other study has observed similar observation. This section was written likes a product insert which could be lack of scientific and critical review/inputs from literature.

R: It has been thoroughly revised.

  1.       Several flow charts or figures in relevant sections may help the readers to understand better.

R: The flow chart has been added and two figures also (Fig 1 and 2).

  1. Line – 167 “Additionally, monitoring the stability of the activimeters' response must be carried out for each channel used to detect any potential drift.” – Kindly elaborate how this should be done?

R: It has been elaborated and shown in line 124.

  1. Section 2.2 should be repositioned to Section 3, a stand-alone section

R: Sections 2.2 and 3 have been thoroughly revised. Some elements of section 2.2 are included in section 3.

  1. The “+/-“ should be replaced with symbol

R: the changes have been made, using the “±” symbol.

  1. Missing Title for table 1 and table 2.

R: the change has been made.

  1. Abbreviation used in the table should be defined in footnote.

R: the change has been made.

  1. Section 3.1.1 and 3.1.2 should be rewritten in paragraph format rather than “XXXX :” Alternative these point forms should be converted to graphic.

R: the section 3.1 has been largely remained, rewritten in paragraph format.

  1. The discussion may need more critical rewrite in such whats the take home message for the readers? whats the gap of knowledge? whats authors suggestion for next actions for the way forward.

R: Thank you for this insightful suggestion. I would like to express as ideas:

  • The evidence base must be strengthened. Dosimetry, with multiple SPECT/CT scans, is unlikely to become a standard practice in clinical settings
  • We need firstly to incorporate dosimetry into clinical trials more frequently in order to show more correlations, if they exist, with the relevant clinical outcomes like PFS or OS.
  • We should improve the practice of post-treatment dosimetry with the streamlined single SPECT/CT scan schemes, as the total integrated activity (and thus radiation dose) can be estimated from a single time sample.
  • The ideal way to overcome the dosimetry based on the post therapeutic imaging is to develop the dosimetry based on the pretherapeutic imaging.

  1. Abbreviation should be defined at first use, example but not limited to:
  2. Line 65 – define EANM/SNMMI
  3. Line 68 – define ANSM
  4. Line 90 – define RLT

R: the changes have been made.

Round 2

Reviewer 2 Report

Comments and Suggestions for Authors

All of my prior concerns have been addressed

Reviewer 3 Report

Comments and Suggestions for Authors

Authors have addressed the concerns and the paper is ready for publication.

Comments on the Quality of English Language

Acceptable English proficiency